# Real-time observation of the buildup of polaron in α-FAPbI₃

Xingyu Yue[1,2,3,6], Chunwei Wang[1,3,4,6], Bo Zhang[5,6], Zeyu Zhang ✉[1,2,3], Zhuang Xiong[5], Xinzhi Zu[1,2,3], Zhengzheng Liu[1,2,3], Zhiping Hu[2], George Omololu Odunmbaku[5], Yujie Zheng[5], Kuan Sun ✉[5] & Juan Du ✉[1,2,3]

The formation of polaron, i.e., the strong coupling process between the carrier and lattice, is considered to play a crucial role in benefiting the photoelectric performance of hybrid organic-inorganic halide perovskites. However, direct observation of the dynamical formation of polarons occurring at time scales within hundreds of femtoseconds remains a technical challenge. Here, by terahertz emission spectroscopy, we demonstrate the real-time observation of polaron formation process in FAPbI₃ films. Two different polaron resonances interpreted with the anharmonic coupling emission model have been studied: P1 at ~1 THz relates to the inorganic sublattice vibration mode and the P2 at ~0.4 THz peak relates to the FA⁺ cation rotation mode. Moreover, P2 could be further strengthened than P1 by pumping the hot carriers to the higher sub-conduction band. Our observations could open a door for THz emission spectroscopy to be a powerful tool in studying polaron formation dynamics in perovskites.

Over the past decade, the photoelectric conversion efficiency (PCE) of halide perovskite photovoltaic devices have displayed a remarkable increase, advancing to 25.6% on black-phase formamidinium lead iodide (α-FAPbI₃)[1]. The superior photoelectric properties of HOIPs can be attributed to the formation of large polarons, which can protect the carriers from recombination centers and defects[2,3], resulting in relatively long carrier lifetime and diffusion lengths[4–6]. Polarons are quasiparticles formed by coulomb interaction between local excess carriers (electron or hole) and surrounding lattice distortions[7]. Based on the soft inorganic sublattices and organic cation disorder of ABX₃-type HOIPs[3,8], it has been reported that the coupling of PB-X vibration modes and carrier electronic motion should be considered to determine polaron formation and carrier transport[9,10]. Traditionally, polaron correlations are investigated by the detection of phonon induced resonance by means of infrared absorption spectroscopy.

Several time-resolved multi-THz spectroscopy (TRTS) results have identified longitudinal optical (LO) phonon modes in HOIPs with resonance frequencies 1–5 THz[10,11]. However, TRTS mainly detects the transport process of the carrier with a LO phonon resonance, and is difficult to detect electron-phonon coupling in the polaron formation process. Therefore, it is still challenging to provide solid evidence for the polaron formation process arisen from the strong coupling between the carrier and a given phonon mode in the sub-picosecond time scale.

The terahertz radiation corresponding to the phonon oscillation can be detected in the polar semiconductor under the ground band disturbance of the external electric field[12–14]. Due to the spontaneous polarization and Dember electric field in HOIPs[15,16], one could expect the strong coupling between carrier and ionic lattice can also produce bremsstrahlung radiation during polaron formation which is related to

¹State Key Laboratory of High Field Laser Physics and CAS Center for Excellence in Ultra-intense Laser Science, Shanghai Institute of Optics and Fine Mechanics (SIOM), Chinese Academy of Sciences (CAS), Shanghai 201800, China. ²School of Physics and Optoelectronic Engineering, Hangzhou Institute for Advanced Study, University of Chinese Academy of Sciences, Hangzhou 310024, China. ³Center of Materials Science and Optoelectronics Engineering, University of Chinese Academy of Sciences, Beijing 100049, China. ⁴School of Physical Science and Technology, ShanghaiTech University, Shanghai 201210, China. ⁵MOE Key Laboratory of Low-grade Energy Utilization Technologies and Systems, CQU-NUS Renewable Energy Materials & Devices Joint Laboratory, School of Energy & Power Engineering, Chongqing University, Chongqing 400044, China. ⁶These authors contributed equally: Xingyu Yue, Chunwei Wang, Bo Zhang. ✉e-mail: zhangzeyu@ucas.ac.cn; kuan.sun@cqu.edu.cn; dujuan@siom.ac.cn

the coherent phonon mode at THz frequency. This provides the possibility of studying the dynamics of polaron formation in HOIPs using THz emission spectroscopy. In addition, A-site organic cations are believed to be able to position carriers by dipole interaction at the initial stage of polaron formation[17], but whether there is a strong coupling between organic cation rotation and carriers is still an open question. Here, the photo-induced polaron formation in black-phase FAPbI3 has been evidenced by real-time terahertz emission spectroscopy. The photo-induced Dember current and the anharmonic oscillator radiation of the polaron modes in FAPbI3 films have been investigated. Two identical polaron modes corresponding to A-site cation rotation ~0.4 THz and inorganic sub-lattice vibration ~1 THz, have been revealed respectively. At pump photon energies above 2.4 eV, the ~0.4 THz mode emission is enhanced, and is related to the A-site cation collision with the excited hot carriers in the upper split conduction band containing FA+ energy level. Our results have not only proved terahertz emission spectroscopy to be a novel experimental technique to observe the polaron formation dynamics, but also revealed a new relaxation channel for the hot carriers involved in polaron formation in HOIPs.

## Result and discussion
### The Dember current THz emission in HOIPs
The lead halide perovskite thin films of were fabricated on quartz via two-step method, as detailed in Methods. Two types of films studied here are marked as the following: 0.95FA for $(FAPbI_3)_{0.95}(MAPbI_3)_{0.05}$ and 0.85FA for $(FAPbI_3)_{0.85}(MAPbBr_3)_{0.15}$. Figure S1 shows the XRD data for 0.85FA and 0.95FA, the bromine content leads to the increase of the diffraction angle of the diffraction peak[18]. THz radiation provides a direct measurement of picosecond time scale photocurrent within the HOIPs film, as shown in Fig. 1a. Under femtosecond laser excitation (35 fs, 1 kHz), the transient photocurrent $J$ radiates a coherent THz radiation from the surface of the HOIPs film. The emitted THz amplitude and phase are detected by free space phase-sensitive

electro-optic sampling in the transmission direction. Here, the THz emission related to the transient photocurrent is investigated by changing the incident angle $\theta$ and polarization direction of the pump light together with the azimuth angle $\Phi$ of the sample.

As shown in Fig. 1b, the time domain terahertz emission signals contain a strong main pulse and a periodically modulated oscillation in several picoseconds. The inset shows that the THz amplitude is linearly depended on laser fluence from 4.7 to 93.8 μJ/cm² Therefore, the observed THz emission is possibly resulting from the transient photogenerated current[19–23], rather than higher-order nonlinear effects[24]. If the THz is generated by transient photocurrent radiation, it will be dependent on the incident angle of the pump light and could be described as follows[25,26]

$$E_{x'}^{THz} \propto t_p \left( \cos\theta_{THz} \frac{\partial J_x}{\partial t} + \sin\theta_{THz} \frac{\partial J_z}{\partial t} \right) \quad (1)$$

$$E_{y'}^{THz} \propto t_s \frac{\partial J_y}{\partial t} \quad (2)$$

where $E_{x'}^{THz}$ and $E_{y'}^{THz}$ are the radiating THz electric field components perpendicular to each other; $t_s$ and $t_p$ are transmittances of S-polarized light and P-polarized light respectively; $J_x$, $J_y$ and $J_z$ are currents in the X, Y and Z directions respectively. Figure S5 shows the THz amplitude depending on the angle of the THz polarizer. The THz amplitude reaches the maximum at $\varphi = 0°$ and $\varphi = 180°$ and zero at $\varphi = 90°$, indicating the emitted THz field is parallel to the incidence plane (p-polarized). From the pump incidence angle dependent terahertz emission, as shown in Fig. 1c, the experimental results can be well fitted by only the second term of Eq. (1). Thus, it can be concluded that the net photocurrent is perpendicular to the material surface. Figure 2a illustrates the generation of THz radiation by the ultrafast transient photocurrent normal to the sample surface, known as the photo-Dember effect[23]. Upon excitation, photo-induced carriers

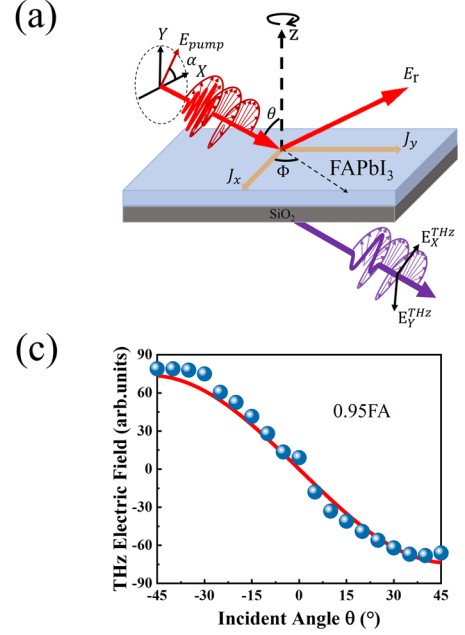
(a)

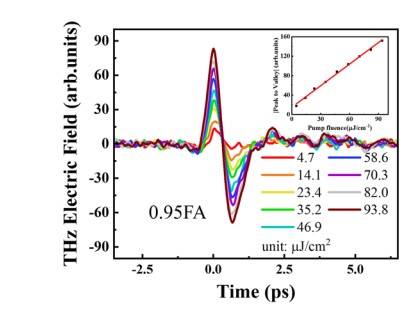
(b)

(c)

(d)
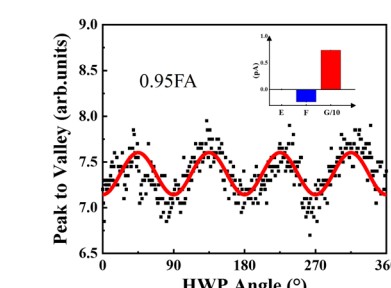

**Fig. 1 | The Photo-induced THz emission in FAPbI3. a** Transient photocurrent distribution and induced THz radiation of HOIPs thin film under femtosecond (fs) laser excitation. *xyz* and *XYZ* represent the crystal and laboratory coordinate. θ is incident angle, *Φ* is the azimuth angle of sample, and $J_x$, $J_y$ and $J_z$ are the photocurrent components generated by pump light irradiation on sample, respectively; **b** THz time-domain emission spectra of 0.95FA measured with different excitation fluence at incident angle of 45°; Inset in (**b**) are pump fluence dependent peak-to-valley value of THz electric field amplitude with 480 nm laser excitation; **c** Incident angle θ dependent peak-to-valley value of THz electric field amplitude with 480 nm laser excitation, the fitted red line through the data points is based on the second term of Eq. (1); **d** THz electric field amplitude dependent on the polarization direction of pump light in 0.95FA, inset are the fitting parameters.

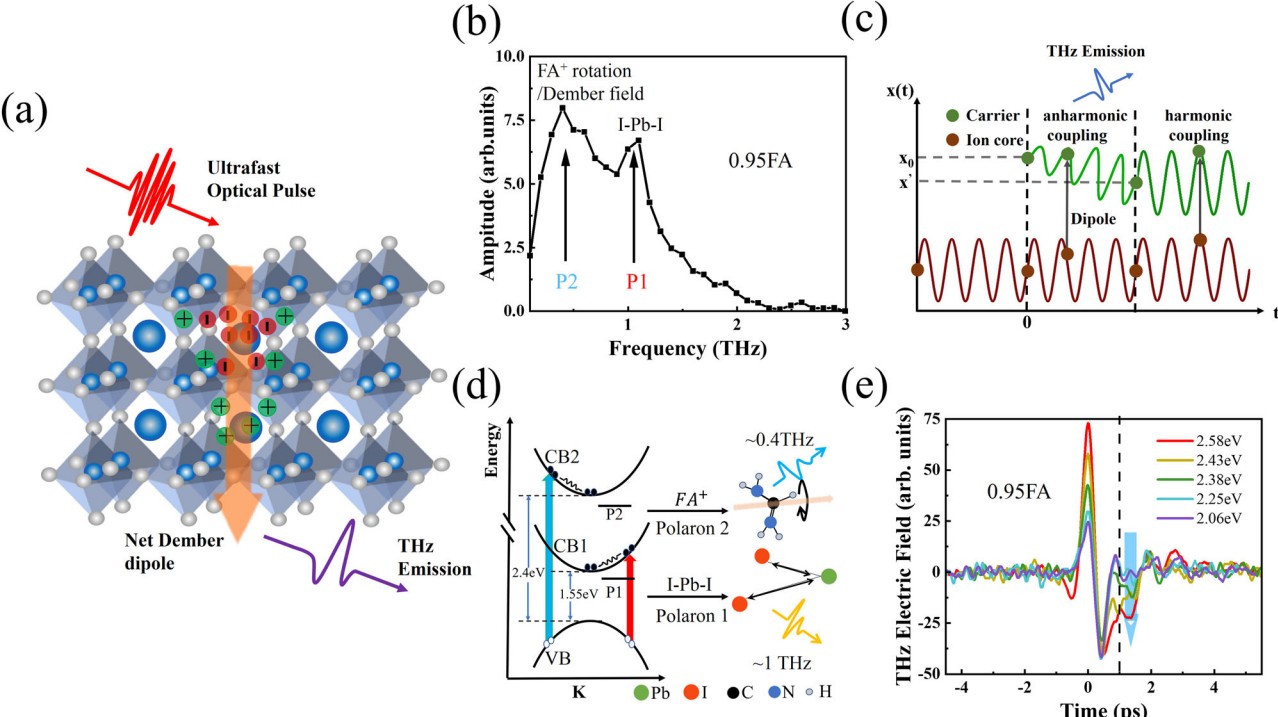

**Fig. 2 | The polaron emission mechanism in FAPbI₃. a** Schematic illustration of the THz emission from the photo-Dember effect. **b** Amplitude spectrum of the emitted THz field form 0.95FA exhibiting two distinct peaks at -0.4 and -1 THz, excited by 2.58 eV pump photon energy; **c** Schematic of THz emission resulting from polaron formation, $X_0$ and $X'$ are the distance of the carrier with respect to the ion, the coupling between carrier and ionic lattice goes through two stages: anharmonic coupling and harmonic coupling. **d** Schematic of FAPbI₃ energy band structure, CB1 corresponds to PB-I energy level, CB2 corresponds to FA⁺ energy level, red and blue arrows correspond to the transition process of VB-CB1 and VB-CB2; **e** THz time-domain emission spectra of 0.95FA excited by different pump photon energies (carrier concentration $n = 1.25 \times 10^{18}$ cm⁻³).

diffuse perpendicularly through the sample under a concentration gradient. Due to the mobility difference of electrons and holes in HOIPs[16,27], the electrons and holes become spatially separated and form a Dember electric field. Under the modulation of the diffusion and Dember electric field, the excited electrons and holes would form a rapidly changing dipole and emit THz radiation. Analogically, different polarized light will lead to photocurrent in different directions. A half-wave plate was used to change the polarization of the pump light. As shown in Fig. 1d, the THz electric field shows a slight periodic change with the rotation of the half-wave plate, which could be caused by the absorption difference of the *S* and *P* light. Meanwhile, ultrafast photoinduced shift currents can also be generated in semiconductors through circular photogalvanic effect (CPGE)[20] and linear photogalvanic effect (LPGE)[28]. The CPGE and LPGE in HOIPs were discussed in detail in Supplementary Note S1, and analysis of the results showed that the THz radiation caused by CPGE and LPGE was relatively weak. As shown in Fig. S4, the thickness of 0.95FA and 0.85FA are -658 nm and -534 nm, respectively. The penetration depth of the pump pulse is taken to be about 1 μm[29,30], which is larger than the thicknesses of the samples. Therefore, the dominant contribution to the observed terahertz pulse in our experiment is attributed to the photo-Dember current origin from the bulk effect.

## The anharmonic coupling polaron emission mechanism in FAPbI₃

The frequency domain spectroscopy of the emitted THz with a pump photon energy at 2.58 eV was obtained by fast Fourier transform. As shown in Fig. 2b, two resonant peaks located at -0.4 THz and -1 THz could be observed. It is reported that the spectral resonance at -1 THz coincides with a LO phonon mode associated with I-Pb-I angular distortions of the inorganic sublattice in FAPbI₃[9,16], while the 0.3-0.5 THz resonant peak could be attributed to the FA⁺ cation rotational

mode[31,32]. The excitation of coherent phonons caused by transient coherent pulse laser has been widely studied[12,14]. From semiclassical electromagnetism, it is not possible to extract a net power from damped charged harmonic oscillators, however with external electric field, the electromagnetic radiation corresponding to the phonon oscillation can be detected in the polarized semiconductor[33,34]. HOIPs is a system with spontaneous ferroelectric polarization[35,36], hence the interaction between the polar lattice and the photoinduced Dember current could radiate the electromagnetic wave corresponding to the phonon oscillation, which makes it possible to use the corresponding electromagnetic emission (here THz emission) to detect the real-time process of carriers coupled to the phonon mode, i. e. the transient formation process of polaron. Although the vibrational phonon modes in HOIPs could be performed using harmonic approximation for the potential energy surface[37], the coupling between the vibration and the photo Dember current could be anharmonic due to their initial frequency discrepancy. As shown in Fig. 2c, we introduce the anharmonic coupling model to demonstrate that the polaron THz emission corresponds to a phonon mode with strong electron-phonon coupling between Dember current and ionic lattice. Before photoexcitation, the ionic lattice and organic cation collectively vibrate near the equilibrium position determined by minimization of the free energy of the system. After photoexcitation, the carriers exhibit two kinds of motions. Firstly, carrier approach or move away from the ionic lattice and organic cation and reach to a new equilibrium position; Secondly, the carriers are coupled to the vibrational or rotational mode of the ionic lattice and A-site cation, respectively. The two kinds of motion cause the photogenerated carriers to undergo an anharmonic vibration, and a rapidly changing dipole moment is formed between the ion core and the carrier, thus emitting a THz signal corresponding to the specific phonon mode, which is also consisted with the bremsstrahlung radiation due to the hot carrier deceleration and energy

reduction. The anharmonic coupling will lead to bremsstrahlung associated with terahertz phonon, resulting in carriers relaxing into metastable states (polarons) through interaction with phonon in momentum space, as shown in Fig. 2d. As shown in Fig. S11b, transient optical conductivity acquired at 5 ps and 100 ps delay times after the photoexcitation also shows two resonance peaks at -0.4 THz (P2) and -1 THz (P1), indicating the relatively long transportation lifetime for the metastable polarons[29,38]. Therefore, the duration of the photo-current oscillation is corresponding to anharmonic coupling time between the Dember current and the lattice vibration, and hence could be connected with the polarons formation time. As shown in Fig. 2e, the periodically modulated oscillation in the time domain after 1 ps represented the polaron formation process in HOIPs, which lasts for about 3 ps.

The hot carriers with excess energy in HOIPs were generally considered to be coupled to inorganic sublattice vibration (-1 THz) after fast relaxation to the bottom of conduction band[9,39]. Meanwhile, it has been reported theoretically that there are two different conduction bands (CB1 and CB2) existing in FAPbI$_3$, and the energy gap from the top of the valence band to FA cation level is about 2.4eV[40–42]. Therefore, it is reasonable to speculate whether the hot electron could couple with the A-site cation rotation (-0.4 THz). As shown in the Fig. 2d, red and blue arrows correspond to the transition process of VB-CB1 and VB-CB2, respectively. According to the DFT simulations presented in the Fig. S13, with the existence of MA substitution, the transition is still direct transitions from VB to CB1 and CB2 bands. To explore whether the physical process corresponding to the peak near -0.4 THz is related to A-site cation rotation, different pump photon energies were employed to excite the films. As shown in Fig. 2e, THz field amplitude in main peak increases with excitation photon energies at constant carrier density $n = 1.25 \times 10^{18}$/cm$^3$. Since hot carriers exhibit higher initial temperature, this leads to enhanced charge separation and the build-up of a transient Dember field[43]. Meanwhile, the THz time-domain waveform changed at 0.5 ps delay time when excited photon energy is above 2.43 eV, indicating a change in the emission mode frequency. Figure 3a depicts the amplitude spectrum of the emitted field, the -0.4 THz peak increased steeply with the photon energy change from 2.38 eV to 2.43 eV, and the amplitude of the -0.4 THz peak in 0.95FA surpasses that of the -1 THz peak when the pump photon energy is higher than 2.4 eV. It has been reported that FA cation rotational phonon mode is in the frequency of -0.3–0.5 THz[31,32] and the FA cation energy level is involved only in the higher sub-conduction band in HOIP[40–42]. Therefore, it is reasonable to assign the enhanced part of THz emission at P2 to the collision between rotated FA cation and the hot electrons with increased density in the FA energy level. This indicates that there are two different polaron modes in the

THz emission spectrum. One is the polaron mode dominated by the vibration of the inorganic sublattice, and the other is the polaron mode with the participation of the A-site cation, which are respectively labeled as polaron 1 (P1) and polaron 2 (P2). The existence of polaron P2 open a new way for the hot carrier induced polaron formation in FAPbI$_3$, which is crucial for improving the utilization of excess energy in perovskite solar cells. In addition, the selective excitation of A-site cation rotation and carrier coupling mode further proves the anharmonic coupling process of the polaron terahertz emission. With excited photon energy higher than 2.4 eV (CB2), the large increase in the amplitude of mode P1 might be caused by the interaction of cationic rotation and the Pb-I vibration. After the hot carriers being pumped above the bottom of CB2, the excess energy could be employed to strengthen the cationic rotation and hence the interaction between cationic rotation and the Pb-I vibration[44].

## The substitution effect on the polaron emission

Finally, in order to study the influence of the A-site cations substitution on the dynamics of P2 mode in HOIPs, the -0.4 THz emission have been compared between 0.85FA and 0.95FA, as shown in Fig. 3b. The detailed amplitude spectrum of the emitted THz field with different pump photon energies are shown in Fig. S 8. Similarly, the amplitude of the P2 mode in 0.85FA surpasses that of the P1 mode when the pump photon energy is higher than 2.38 eV. As shown in Fig. 3c, the peak amplitude of 0.95FA and 0.85FA at P2 mode change with excitation photon energy. The different growth rate of P2 mode with photon energy for the two samples can be explained in the following way. The MA cation have higher dipole moment than FA in a cubic phase perovskite lattice at room temperature[3,45]. Then, the MA rotation intensity is expected to increase easier than the FA with the collisions of the hot carriers. This assumption has been confirmed by further experiments in the samples with different proportion of the MA cation (MAPbI$_3$, (FAPbI$_3$)$_{0.85}$(MAPbI$_3$)$_{0.15}$, (FAPbI$_3$)$_{0.95}$(MAPbI$_3$)$_{0.05}$ and FAPbI$_3$). The photon energy dependence of the P2 mode terahertz emission have been compared. As shown in Fig. S9, the P2 increases most significantly in the MAPbI$_3$ sample which has the largest proportion of MA cations, and vice versa. These results further confirm the terahertz emission spectroscopy could reveal the strong coupling dynamics between the hot carriers and different modes of phonons.

In conclusion, the dynamical polaron formation induced terahertz emission has been real-time demonstrated in black-phase formamidinium lead iodide perovskite. Two polaron modes were resolved by the emitted spectrum under variable pump photon energies. Polaron P1 is in connection with the vibrational mode of Pb-I coupled with the conduction band bottom carrier, and the other polaron P2 is formed by the strong electron-phonon coupling between

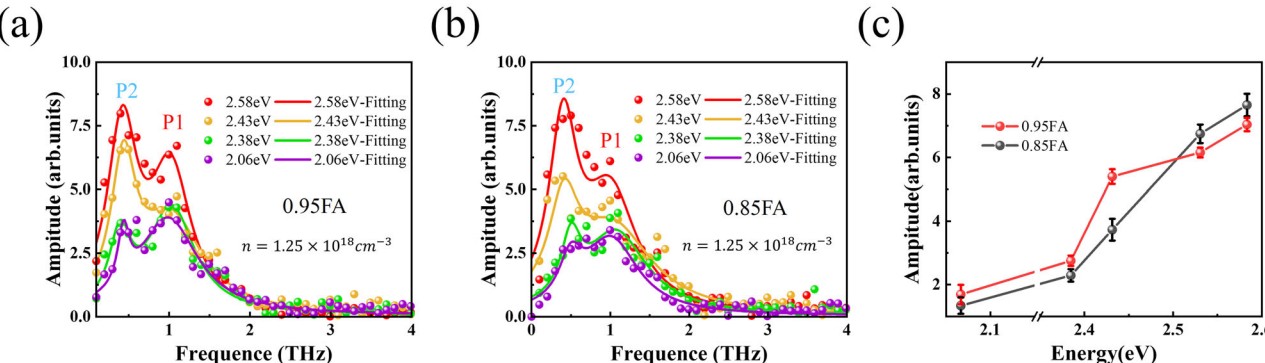

**Fig. 3 | The substitution effect on the polaron emission modes. a** Amplitude spectrum of the emitted THz field form 0.95FA exhibiting two distinct peaks at -0.4 and -1 THz, the solid line uses Lorentz fitting; **b** Amplitude spectrum of the emitted THz field from 0.85FA exhibiting two distinct peaks at -0.4 and -1 THz, the solid line uses Lorentz fitting; **c** The peak values of polaron 2 depend on different excitation photon energies for 0.95FA and 0.85FA.

hot carriers excited to CB2 and the A-site cation rotation mode. The new P2 polaron modes in HOIPs could provide a new channel for utilizing the excess energy in perovskite solar cells. Furthermore, the substitution of A-site organic cations is shown to be able to affect the electron-phonon coupling strength through the dipole moment of the A cation. Our results have not only real-time observed the polaron formation process in HOIPs but also demonstrate a new A-site cation correlated polaron modes that might benefit the hot carrier utilization in HOIPs based solar cells.

## Methods

### Sample fabrication

The precursor $PbI_2$ (691.5 mg, 1.5 mmol/mL for 0.95FA sample and 599 mg, 1.3 mmol/mL for 0.85FA sample) in DMF: DMSO (9:1) solution was spin coated onto the cooled substrate at 1500 rpm for 25 s and 2500 rpm for 5 s, and then annealed at 60 °C for 2 min. After the $PbI_2$ film cooled down to room temperature, 60 μL of the organic mixture solution of FAI: MAI: MACl (91:6.4:9.5 mg for 0.95FA sample in 1 mL IPA) and FAI:MAI:MABr (70:7:7.5 mg for 0.85FA sample in 1 mL IPA) was spin coated onto the $PbI_2$ at 2500 rpm for 3 s, 1350 rpm for 15 s and then 1700 rpm for 12 s, When the resulting film turned from orange to dark brown in glove box, they were thermally annealed at 150 °C for 15 min under ambient condition.

### Terahertz emission spectroscopy

The fundamental laser pulse with wavelength at 800 nm is generated by a Ti:sapphire amplifier. The pulse repetition rate was 1 kHz and the pulse width was 35 fs. The fundamental laser pulse is divided into two beams, one is wavelength-tuned through OPA as pump light, and the other beam is used as probe light to detect THz signal emitted by the sample. Perovskite samples were placed on a rotating sample holder and the THz signal generated by the sample was collected by Wollaston prism.

## Data availability

The raw data that support the findings of this study have been depositedsentence ca in Science Data Bank [Science DB] with the accession codes [NJZbya]; https://doi.org/10.57760/sciencedb.02698.

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

## Acknowledgements

This work was supported by the National Natural Science Foundation of China (Nos. 92050203, 61905264, 62205081, 61875211, 61674023, 62005296, 12004057, and 62074022), the National Key R&D Program of China (2022YFA1604403), and the CAS Interdisciplinary Innovation Team, Hangzhou Science and Technology Bureau of Zhejiang Province (No. TD2020002), Shanghai Pilot Program for Basic Research (22JC1403200).

## Author contributions

Z.Z., J.D., and K.S. conceived of the research; B.Z., Z.X., G.O.O. grew the films; Z. Z., X.Y., C.W., X.Z., Z.L. and Z.H. performed the measurements; Y.Z. performed the theoretical simulation; X.Y., Z.Z. and J.D. wrote the manuscript; all authors commented the manuscript and J.D. led the project.

## Competing interests

The authors declare no competing interests.
