## [Peer Review File · Nature Communications]

Real-Time Observation of the Buildup of Polaron in α -FAPbI₃REVIEWER COMMENTS

Reviewer #1 (Remarks to the Author):

see attachment

In this manuscript, the authors observed two terahertz emission modes from photoexcited FAPbI₃ film and attributed to coupled electron-phonon modes for the large polaron formation process. The topic and the observation of coherent phonon emission is interesting but the interpretation and conclusion of this work is highly problematic, which prevent the publication of this manuscript. Below are details:

Major concerns:

The authors observed phonon emission and attributed to photo-Dember current dressed by polar lattice vibration. It is reasonable until here. But I did not see any correlation between these phonon modes and large polaron formation. For example, I don't understand the key statements below

...“As a strong electron-phonon coupling system with ferroelectricity, the strong electron-phonon coupling in HIOPs should be expected the THz radiation corresponding to the large polaron mode.”...

...“Therefore, the duration of the anharmonic oscillation of the carriers corresponds to the time of polaron formation.”...

1. In fact, the authors use 480nm laser excitation which is way above the band edge, the cooling of hot carriers can also generate phonon emissions to dump the excess energy. It does not necessarily involve large polaron at all.

2. Even at band edge excitation, the coherent phonon emission can be generated as long as the pulse is short enough and the potential energy surface at excited state is displaced relative to ground state. (*Nat. Commun.* **2018**, *9* (1), 2525) These modes could be related to polaronic relaxation but has nothing to do with large polaron mode. To claim THz from large polaron mode, the authors must show some phonon frequency shifting corresponding to the structural distortion, for example.

3. The assignment of oscillation duration to the large polaron formation time is wrong. The decay of these phonon oscillation should be mostly due to phonon-phonon coupling (i.e. phonon anharmonicity). (*J. Am. Chem. Soc.* **2020**, *142* (39), 16569)

4. The authors talked about “anharmonic oscillation” multiple times all through the manuscript. In addition to the statement above, see also

...“The selective excitation of A-site cation rotation and carrier coupling mode further proves the anharmonic oscillation process of the polaron terahertz emission”...

Anharmonic indeed is important to this family of materials. But from results, I did not see evidence or signature of anharmonic. Everything can be explained with in the harmonic potential. Please clarify.

Minor concerns:

1. Fig. 3b inset, why there is an intercept for the linear fitting.
2. The authors mention two samples in the manuscript, 0.95FA and 0.85FA. It is confusing that 0.85FA is indicated as (FAPbI₃)_{0.85}(MAPbI₃)_{0.15} in the discussion section but (FAPbI₃)_{0.85}(MAPbBr₃)_{0.15} in Fig S1. The author says “the increase of bromine content”. I have no idea about what's the real composition of the sample.
3. P2 mode of 0.85FA sample appears redshifted with the enhancement of the incident photon energy but no shift is found in 0.95FA. What's the explanation?

4. The P1 intensity is almost the same at 2.06eV, 2.38eV and 2.43eV excitation, but there is a large increase at 2.58eV excitation. What's the reason?
5. What is the specific reason for the different growth rate of P2 mode with photon energy for the two samples?

Reviewer #2 (Remarks to the Author):

This manuscript reported the real time observation of the polaron formation process in FAPbI₃ solar cells by terahertz emission spectroscopy with bringing up a phenomenological model to describe the electromagnetic radiation from the polar lattice and carriers at non-equilibrium state. Interestingly, the controlling over of the polaron 2 consisted with the FA⁺ cation rotation mode at 0.45 THz peak also conform to the anharmonic oscillator emission of the large polaron formation with a photon excitation energy higher than 2.4 eV. Overall, it is an original and interesting work, and their efforts will give insight into the dynamical transportation for large polarons in perovskite solar cells. Therefore, I would recommend the publication of this paper in Nature Communications after required revision. The following comments and questions need to be considered.

1. The thickness of the perovskite samples should be presented to distinguish the transient current origin of the bulk and the surface origin.
2. The THz time domain spectroscopies and optical pump-THz probes of the perovskite films could help to distinguish the polaron mode and the phonon mode.
3. At the end of the manuscript, the comparison of the FA_{0.85} and the FA_{0.95}, what is the physical insight of this comparison especially on the P2 polaron mode?
4. Please comment that what are the mobilities of photo-excited electrons and holes in the samples in the paper.
5. The English grammatical errors that need to be corrected by the authors throughout their manuscript.

Reviewer #3 (Remarks to the Author):

This study opens a new door for THz emission spectroscopy to be a powerful tool in studying polaron formation in HOP by observing anharmonic oscillator emission of THz emission. I recommend this manuscript can be accepted after the following minor comments are addressed.

Minor comments

1. It is not clear from the manuscript why the authors use different element substitution modification systems of FAPbI₃ (0.95FA and 0.85FA) rather than FAPbI₃ itself. Is there any physical or chemical reason for this? According to the Fröhlich coupling constant, a large strength can be expected if the effective mass of the charge carrier is large and/or the angular frequency of the coupled LO phonon is smaller. Do the authors intend to observe any effects of the substitution on the electron-lattice coupling? Or are

there any DFT simulations that indicate any difference in the effective mass or LO phonons by the substitution modification?

I am interested in the case where FAPbI₃ is used for the same experiment. The result would be a good reference.

2. A model based on FAPbI₃ is presented in the manuscript (Fig 2 (d)). I wonder how this model (vertical electronic excitations CB1 and CB2) can be justified under the existence of MAPbI₃. If the authors simply assume this, then is there any DFT simulation method that can support this model? There are several ways to perform DFT simulations to justify this type of model: for instance, the finite size of the structure can be used to calculate both the electronic ground state and the electronically excited state energy along a relevant I-Pb-I normal coordinate (or a simple motion) to see if the equilibrium structure along this coordinate changes. Or the orientation of the FA⁺ molecule can be altered along with the corresponding normal coordinate (or a simple motion) and the electronic energy can be simulated at each configuration. If the authors can provide these pieces of information, this model can be more convincing.

3. Electronic excitation can take place at any K-points within the allowed energy window creating electrons and electron holes with each pair of electron-hole having opposite crystal momenta. How is the crystal momentum of phonon involved in Fig. 2(d)? Do the authors assume that THz emission does not need any crystal momentum of the relevant THz phonon? Or if the authors focus only on the Gamma point or specific k-point, please mention it in the manuscript.

4. Related to comments 1 & 2, can the authors identify the origins of P1 and P2 peaks of 0.95FA are the same as those of MAPbI₃ or FA_{0.83}Cs_{0.17}PbBr_{0.51}I_{2.48} or FAPbI₃? Is it possible for the authors to provide more specific information on the relevant THz modes of 0.95FA (and 0.85FA) and the effects of substitution modification on the modes?

Response to The Reviewers' Concerns:

Reviewer #1:

In this manuscript, the authors observed two terahertz emission modes from photoexcited FAPbI₃ film and attributed to coupled electron-phonon modes for the large polaron formation process. The topic and the observation of coherent phonon emission is interesting but the interpretation and conclusion of this work is highly problematic, which prevent the publication of this manuscript. Below are details:

Response: Thanks for your positive comments on our research topic and observation. About the interpretation and conclusion part, we are sorry for the misunderstanding due to unclear descriptions in our previous version. We have made corresponding modifications according to your comments.

Major concerns:

The authors observed phonon emission and attributed to photo-Dember current dressed by polar lattice vibration. It is reasonable until here. But I did not see any correlation between these phonon modes and large polaron formation. For example, I don't understand the key statements below

... "As a strong electron-phonon coupling system with ferroelectricity, the strong electron-phonon coupling in HOIPs should be expected the THz radiation corresponding to the large polaron mode." ...

... "Therefore, the duration of the anharmonic oscillation of the carriers corresponds to the time of polaron formation." ...

Response: We are sorry for the misunderstanding due to our unclear description. We have corrected the descriptions as "HOIPs is a system with spontaneous ferroelectric polarization [Science advances 5, eaas9311 (2019); Advanced Materials 31, 1806661 (2019)], hence the interaction between the polar lattice and the photoinduced Dember current could radiate the electromagnetic wave corresponding to the phonon oscillation, which makes it possible to use the corresponding electromagnetic emission (here THz emission) to detect the real-time process of carriers coupled to the phonon mode, i. e. the transient formation process of polaron." "Therefore, the duration of the photo-current oscillation is corresponding to anharmonic coupling time between the Dember

current and the lattice vibration, and hence could be connected with the polarons formation time.”

The explanation for the first sentence above: From semiclassical electromagnetism, it is not possible to extract a net power from damped charged harmonic oscillators, however with external electric field, the electromagnetic radiation corresponding to the phonon oscillation can be detected in the polarized semiconductor. [Physical review letters 94, 057408 (2005); Physical review letters 82, 5140 (1999)]. HOIPs is a system with spontaneous ferroelectric polarization, hence the interaction between the polar lattice and the photoinduced Dember current could radiate the electromagnetic wave corresponding to the phonon oscillation, which makes it possible to use the corresponding electromagnetic emission (here THz emission) to detect the real-time process of carriers coupled to the phonon mode, i. e. the transient formation process of polaron. We have added the above-mentioned descriptions and references in the revised manuscript.

The explanation for the second sentence above: In order to distinguish the phonon behavior and polaron formation, we have performed the optical pump-THz probe (OPTH) spectroscopy, and you can see the details in Responses 1 and 3 as below. To explain the differences between the phonon mode and the polaron mode based on our study, we have introduced the anharmonic coupling model to interpret the physical mechanism of the polaron terahertz emission. In our proposed model, the Dember current could couple with the ionic lattice, and then emit the periodically modulated oscillation in time domain terahertz emission signals which vanishes within several picosecond during the formation process of polarons. The completed time of anharmonic coupling observed here is related to previously reported polaron formation time in the sub-picosecond range [Science, 353(6306), 1409-1413; Advanced Materials, 2018, 30(29): 1707312; Science Advances, 2017, 3(10): e1701469].

About the correlation between phonon modes and the polaron formation, we have explained in the Response to Comment 1 as below.

1. In fact, the authors use 480nm laser excitation which is way above the band edge, the cooling of hot carriers can also generate phonon emissions to dump the excess energy. It does not necessarily involve large polaron at all.

Response: Thanks for the valuable comments. The band edge of CB2 is $\sim 2.4\text{eV}$. According to your comment, we have used photon excitation below (630nm, 1.97 eV) and above (480nm, 2.58 eV) CB2 in the optical pump-THz probe (OPTP) spectroscopy to characterize the photo-induced THz conductivity. In OPTP measurement, the absorption peaks in the photo-induced THz conductivity are related to the intensity of phonon emission [Reviews of Modern Physics 83.2 (2011): 543; Advanced Optical Materials, 2020, 8(3): 1900783]. If the observed result is mainly due to the phonon emissions generated by the cooling of hot carriers associated with the excess energy dumping, different absorption intensity at P2 should be detected using photon excitation below and above CB2. However, as shown in Fig R1, the intensities of P2 are similar. Therefore, it is incomplete to use phonon emissions generated by hot carriers cooling to explain the THz emission enhancement of P2 observed using excitation above bandgap in our study.

The enhancement of THz emission signal at P2 with photon excitation higher than the band edge could be interpreted in the following way. It has been theoretically reported that FA cation rotational phonon mode is in the frequency of $\sim 0.3\text{-}0.5\text{THz}$ [The Journal of Physical Chemistry Letters 6, 3209-3212 (2015); Journal of the American Chemical Society 139, 4068-4074 (2017)], which is in consistent with our P2 signal. In addition, it has been reported that the FA cation energy level is involved only in the higher sub-conduction band in HOIP [New Journal of Chemistry 45, 4393-4400 (2021); Journal of Applied Physics, 2017, 121(11): 115501; Scientific reports, 2021, 11(1): 1-10], which is also in consistent with our enhanced P2 signal which increases sharply only at excitation higher than CB2. Therefore, it is reasonable to assign the enhanced part of THz emission at P2 to the collision between rotated FA cation and the hot electrons with increased density in the FA energy level. **We have added the corresponding discriptions in our revised manuscript. (Page 9 and Fig. S12)**

Figure R1. Real part of transient optical conductivity of 0.95FA with delay time of 5ps following photoexcitation at 2.58 eV and 1.97eV with pump fluences of $70.3 \mu\text{Jcm}^{-2}$ at 300 K.

2. *Even at band edge excitation, the coherent phonon emission can be generated as long as the pulse is short enough and the potential energy surface at excited state is displaced relative to ground state. (Nat. Commun. 2018, 9 (1), 2525) These modes could be related to polaronic relaxation but has nothing to do with large polaron mode. To claim THz from large polaron mode, the authors must show some phonon frequency shifting corresponding to the structural distortion, for example.*

Response: Thanks for your comments. It is true that even at band edge excitation, the coherent phonon emission can be generated as long as the pulse is short enough and the potential energy surface at excited state is displaced relative to ground state. But according to the Response to Comment 1 above, it is incomplete to use phonon emission to explain the THz emission enhancement at P2.

Actually, our work focused on utilizing the terahertz emission spectroscopy to real-time characterize the polaron formation process in HOIPs at ultrafast time scale. The description of the “large polaron” in our manuscript comes from the mobility decreasing behavior during the polaron formation process. Based on your comments, we recognize that it is not sufficient to determine whether the polaron mode is large or small just according to one reason. Anyway, the spatial scale of the polarons is not the main focus of our work. No matter it is small or large, it will not affect our key point that THz emission spectroscopy could be a powerful tool in studying polaron formation

in HOIPs. **Therefore, we replaced the “large polaron” by “polaron” in our revised manuscript.**

3. The assignment of oscillation duration to the large polaron formation time is wrong. The decay of these phonon oscillation should be mostly due to phonon-phonon coupling (i.e. phonon anharmonicity). (J. Am. Chem. Soc. 2020, 142 (39), 16569)

Response: Thanks for the valuable comments. To check the relaxation time of phonon mode in HOIPs, we have performed the OPTP spectroscopy. As shown in Fig.R2, the peak intensity of two phonon modes at delay time of 100ps only decreases slightly compared with the one at delay time of 5ps, which indicates the lifetime of phonon modes is much longer than 100ps. This is also considered to be the lifetime of metastable polarons in literatures. [Light: Science & Applications, 2022, 11(1): 1-12; Physical review letters, 2019, 122(16): 166601; ACS Energy Lett. 2021, 6, 2, 568–573; Adv. Mater. 2018, 30, 170731] However, the THz emission vanishes within several picoseconds. Therefore, the phonon-phonon coupling or the phonon anharmonicity could not be the reason for decay of oscillation observed in THz emission spectroscopy. The THz emission with short lifetime could be explained by the anharmonic coupling model, in which the periodically modulated oscillation in time domain THz emission comes from the anharmonic coupling between Dember current and the ionic lattice. **We have used the description of “anharmonic coupling time between the Dember current and the lattice vibration” instead in our revised manuscript.**

Figure R2. Real part of transient optical conductivity of 0.95FA with delay time of 5ps and 100ps with pump fluences of $70.3 \mu\text{Jcm}^{-2}$ at 300 K.

4. The authors talked about “anharmonic oscillation” multiple times all through the manuscript. In addition to the statement above, see also ... “The selective excitation of A-site cation rotation and carrier coupling mode further proves the anharmonic oscillation process of the polaron terahertz emission”... Anharmonic indeed is important to this family of materials. But from results, I did not see evidence or signature of anharmonic. Everything can be explained with in the harmonic potential. Please clarify.

Response: Thanks for the valuable comments. The anharmonic oscillation mentioned in our model means the anharmonic coupling between the photo Dember current and the ionic lattice, rather than the anharmonic potential for the phonon emission. Indeed, the vibrational phonon modes in MAPbI_3 could be performed using harmonic approximation for the potential energy surface [Nat. Commun. 2018, 9 (1), 2525]. But the coupling between the vibration and the photo Dember current should be anharmonic due to their initial frequency discrepancy. The anharmonic coupling will lead to bremsstrahlung associated with terahertz phonon, resulting in carriers relaxing into

metastable states (polarons) through interaction with phonon in momentum space. **To make our statements more clearly, we have replaced “anharmonic oscillation” by “anharmonic coupling” and added the corresponding reference in our revised manuscript.**

Minor concerns:

1. Fig. 3b inset, why there is an intercept for the linear fitting.

Response: Thanks for the valuable comments. There is no inset figure in our Fig.3. Do you mean the inset of Fig. 1b? This intercept is also observed in other groups' results [Advanced Functional Materials, 2019, 29(40): 1904694; Advanced Materials, 2018, 30(11): 1704737]. There is nearly no interpretation about it. Maybe it is just artifacts during the calculation.

2. The authors mention two samples in the manuscript, 0.95FA and 0.85FA. It is confusing that 0.85FA is indicated as $(\text{FAPbI}_3)_{0.85}(\text{MAPbI}_3)_{0.15}$ in the discussion section but $(\text{FAPbI}_3)_{0.85}(\text{MAPbBr}_3)_{0.15}$ in Fig S1. The author says “the increase of bromine content”. I have no idea about what's the real composition of the sample.

Response: Thanks for the valuable comments. We are very sorry for the typo in our discussion section. The 0.85FA and 0.95FA samples in our manuscript is $(\text{FAPbI}_3)_{0.85}(\text{MAPbBr}_3)_{0.15}$ and $(\text{FAPbI}_3)_{0.95}(\text{MAPbI}_3)_{0.05}$, respectively. It is much more difficult to control the crystallization process of $(\text{FAPbI}_3)_{0.85}(\text{MAPbI}_3)_{0.15}$ than $(\text{FAPbI}_3)_{0.85}(\text{MAPbBr}_3)_{0.15}$, hence, the solar device performances based on $(\text{FAPbI}_3)_{0.85}(\text{MAPbI}_3)_{0.15}$ is hard to control. The reason why we select them to investigate is that they are might the most stabilized ones in two-step prepared perovskite solar cells. [Nat. Photonics 2019, 13 (7), 500; Adv. Mater. 2022, 34 (8), 2106118; Materials 2021, 14 (14), 4054; J. Am. Chem. Soc. 2015, 137 (51); Adv. Mater. 2017, 29 (35), 1606555; Energy Environ. Sci. 2017, 10 (2), 621]. **We have corrected it to be $(\text{FAPbI}_3)_{0.85}(\text{MAPbBr}_3)_{0.15}$ in our revised discussion section.**

3. P2 mode of 0.85FA sample appears redshifted with the enhancement of the incident photon energy but no shift is found in 0.95FA. What's the explanation?

Response: Thanks for the valuable comments. In order to check the redshift phenomenon further, we have done experiments on more samples, such as FAPbI₃, (FAPbI₃)_{0.85}(MAPbI₃)_{0.15} and MAPbI₃, to check the influence of the MA/FA ratio on the P2 Mode. As shown in Fig R3, no obvious shift for the P2 mode could be found in these samples under different pump photon energies. Therefore, the different ratio of the FAPbI₃/MAPbI₃ could not be the reason for the redshift of P2 mode. The slight red shift might arise from the (MAPbBr₃)_{0.15} component in our 0.85FA sample [Journal of the American Chemical Society, 139(11), 4068-4074].

Figure R3. THz emission spectra of FAPbI₃, (FAPbI₃)_{0.95}(MAPbI₃)_{0.05}, (FAPbI₃)_{0.85}(MAPbI₃)_{0.15} and MAPbI₃, measured with carrier concentration $n=2 \times 10^{18} \text{ cm}^{-3}$.

4. The P1 intensity is almost the same at 2.06eV, 2.38eV and 2.43eV excitation, but there is a large increase at 2.58eV excitation. What's the reason?

Response: Thanks for the valuable comments. The P1 mode is the polaron mode related to the LO phonon mode associated with I-Pb-I angular distortions of the inorganic

sublattice. At excitation with photon energy relatively higher than 2.4eV (CB2), the large increase in the intensity of mode P1 might be caused by the interaction of cationic rotation and the Pb-I vibration. After the hot carriers being pumped above the bottom of CB2, the excess energy could be employed to strengthen the cationic rotation and hence the interaction between cationic rotation and the Pb-I vibration [Advanced Energy Materials 6, 1600422 (2016)]. **We have added corresponding descriptions in the revised manuscript in Page 9.**

5. What is the specific reason for the different growth rate of P2 mode with photon energy for the two samples?

Response: Thanks for the valuable comments. The different growth rate of P2 mode with photon energy for the two samples can be explained in the following way. The MA cation have higher dipole moment and then stronger carrier-molecular dipole interaction than FA in a cubic phase perovskite lattice at room-temperature [Science, 2016, 353(6306): 1409-1413; Accounts of chemical research 49, 528-535 (2016)]. Then, the MA rotation intensity is expected to increase easier than the FA with the collisions of the hot carriers. This assumption has been confirmed by further experiments in which samples with different proportion of the MA cation (MAPbI_3 , $(\text{FAPbI}_3)_{0.95}(\text{MAPbI}_3)_{0.05}$, $(\text{FAPbI}_3)_{0.85}(\text{MAPbI}_3)_{0.15}$ and FAPbI_3) have been compared with each other. As shown in Fig.R3, the P2 increases most significantly in the MAPbI_3 sample which has the largest proportion of MA cations, and vice versa. **We have added corresponding descriptions in the revised manuscript in Page 10.**

Reviewer #2:

This manuscript reported the real time observation of the polaron formation process in FAPbI3 solar cells by terahertz emission spectroscopy with bringing up a phenomenological model to describe the electromagnetic radiation from the polar lattice and carriers at non-equilibrium state. Interestingly, the controlling over of the polaron 2 consisted with the FA+ cation rotation mode at 0.45 THz peak also conform to the anharmonic oscillator emission of the polaron formation with a photon excitation

energy higher than 2.4 eV. Overall, it is an original and interesting work, and their efforts will give insight into the dynamical transportation for large polarons in perovskite solar cells. Therefore, I would recommend the publication of this paper in Nature Communications after required revision. The following comments and questions need to be considered.

Response: We thank the reviewer for the positive feedback and valuable suggestions to improve the quality of our work. According to the reviewer's comments, the manuscript has been revised below.

1. The thickness of the perovskite samples should be presented to distinguish the transient current origin of the bulk and the surface origin.

Response: Thanks for the valuable comments. As shown in Fig. R4, we have presented the thickness of 0.95FA and 0.85FA by means of step profilometer measurement, which are ~658nm and ~534nm, respectively. The penetration depth of the pump pulse is taken to be about 1 μ m determined by thin film absorption [Energy & Environmental Science 8, 3700-3707 (2015); Light: Science & Applications 11, 1-12 (2022)], which is larger than the thicknesses of the samples. Therefore, the photoinduced THz radiation is dominated by the bulk origin in our experiment. **We have added the corresponding figure and description in our revised manuscript (Page 5).**

Figure R4 The thickness of 0.95FA and 0.85FA are $\sim 658\text{nm}$ and $\sim 534\text{nm}$, respectively, as determined by profilometry.

2. The THz time domain spectroscopies and optical pump-THz probes of the perovskite films could help to distinguish the polaron mode and the phonon mode.

Response: Thanks for the valuable comments. We first investigated the THz time domain spectroscopy (TDS) of the 0.95FA film, as shown in Fig. R5(a). The real part of the THz conductivity $\sigma(\omega)$ for the 0.95 FA film without optical pump shows the phonon resonance mode at ~ 0.9 THz, which are assigned to the I-Pb-I angular distortions of the inorganic sublattice (P1) [Advanced Materials 30, 1704737 (2018); Science advances 3, e1701217 (2017)]. Then, the optical pump-THz probe (OPTP) spectroscopy of the perovskite film could also help to detect the photoinduced carrier coupled phonon modes in the 0.95FA film. [Physical Review Letters, 2019, 122(16): 166601; Light: Science & Applications, 2022, 11(1): 1-12; Nano Lett. 17, 5402(2017)]. As shown in Fig. R5(b), transient optical conductivity acquired at 5ps and 100ps delay times after the photoexcitation was presented. Two resonance peaks at $\sim 0.4\text{THz}$ and $\sim 1\text{THz}$ could be found, which are corresponding to the A-site cation rotation polaron (P2) and inorganic sublattice vibration polaron (P1), respectively. [Journal of the

American Chemical Society 139, 4068-4074 (2017); *Advanced Materials* 30, 1704737 (2018); *Light: Science & Applications*, 2022, 11(1): 1-12; *Science advances* 3, e1701217 (2017)] The measured polaron frequencies in OPTP are in consistent with our results obtained using terahertz emission spectroscopy. In contrast, the terahertz emission could present the dynamical anharmonic coupling between the photoinduced carriers and the phonon modes during the formation process of polarons, which have been seldom reported in HOIPs. Therefore, these measurements of TDS and OPTP could support our new proposed anharmonic coupling model about the strong electron phonon coupling, which is experimentally proved by our terahertz emission spectroscopy.

Fig.R5. (a) The THz conductivity $\sigma(\omega)$ of 0.95FA without optical pump; (b) Real part of transient optical conductivity of 0.95FA with delay time of 5ps and 100ps with pump fluences of $70.3 \mu\text{Jcm}^{-2}$ at 300 K.

3. At the end of the manuscript, the comparison of the FA0.85 and the FA0.95, what is the physical insight of this comparison especially on the P2 polaron mode?

Response: Thanks for the valuable comments. We want to introduce the different growth rate of P2 mode dependent on photon energy for the two samples. The MA cation have higher dipole moment and then stronger carrier-molecular dipole interaction than FA in a cubic phase perovskite lattice at room-temperature [*Science*, 2016, 353(6306): 1409-1413; *Accounts of chemical research* 49, 528-535 (2016)]. Then,

the MA rotation intensity is expected to increase easier than the FA with the collisions of the hot carriers. This assumption has been confirmed by further experiments in which samples with different proportion of the MA cation (MAPbI_3 , $(\text{FAPbI}_3)_{0.95}(\text{MAPbI}_3)_{0.05}$, $(\text{FAPbI}_3)_{0.85}(\text{MAPbI}_3)_{0.15}$ and FAPbI_3) have been compared with each other. As shown in Fig.R6, the P2 increases most significantly in the MAPbI_3 sample which has the largest proportion of MA cations, and vice versa. **We have added corresponding descriptions in the revised manuscript. (Page 10)**

Figure R6. THz emission spectra of FAPbI_3 , $(\text{FAPbI}_3)_{0.95}(\text{MAPbI}_3)_{0.05}$, $(\text{FAPbI}_3)_{0.85}(\text{MAPbI}_3)_{0.15}$ and MAPbI_3 , measured with carrier concentration $n=2 \times 10^{18} \text{ cm}^{-3}$.

4. Please comment that what are the mobilities of photo-excited electrons and holes in the samples in the paper.

Response: Thanks for the valuable comments. The mobility difference of the photo-excited electrons and holes could help to quantitatively analyze the photo-Dember effect induced terahertz emission in our results. The photocurrent induced by photo-Dember effect is affected by different electron and hole mobility, which could be

described by $J_{Dember} = -e\Delta D \frac{\partial \Delta n}{\partial z}$, where $\Delta D = \frac{k_B}{e}(\mu_h T_h - \mu_e T_e)$ is the diffusivity difference of the electrons and holes, $\frac{\partial \Delta n}{\partial z}$ represents the spatial gradient of the photogenerated carriers along the film thickness, T is the carrier temperature, and μ is the carrier mobility [Advanced Materials, 2018, 30(11): 1704737]. The mobility difference of electrons and holes in 0.95FA and 0.85FA is $9.93 \text{ cm}^2\text{V}^{-1}\text{s}^{-1}$ and $7.34 \text{ cm}^2\text{V}^{-1}\text{s}^{-1}$, respectively. As the total carrier mobility of the 0.95FA and 0.85FA sample is previously reported as $15.07 \text{ cm}^2\text{V}^{-1}\text{s}^{-1}$ and $11.57 \text{ cm}^2\text{V}^{-1}\text{s}^{-1}$, respectively [Journal of Energy Chemistry 76 (2023): 175-180]. Therefore, the respective mobility of the electrons and holes in 0.95FA and 0.85FA can be listed in Table R1.

Table R1. The Mobility Parameters of 0.95FA and 0.85FA at Carrier Density $n = 1.25 \times 10^{18} \text{ cm}^{-3}$

unit ($\text{cm}^2\text{V}^{-1}\text{s}^{-1}$)	$\mu_h - \mu_e$	$\sum \mu$	μ_h	μ_e
0.95FA	9.93	15.07	12.50	2.57
0.85FA	7.34	11.57	9.455	2.115

5. *The English grammatical errors that need to be corrected by the authors throughout their manuscript.*

Response: Thanks for the valuable comments, we have modified our description and polished the revised manuscript.

Reviewer #3:

This study opens a new door for THz emission spectroscopy to be a powerful tool in studying polaron formation in HOP by observing anharmonic oscillator emission of THz emission. I recommend this manuscript can be accepted after the following minor comments are addressed.

Response: We thank the reviewer for the positive feedback and valuable suggestions to improve the quality of our work. According to the reviewer's valuable suggestions, the manuscript has been revised as below.

Minor comments

1. It is not clear from the manuscript why the authors use different element substitution modification systems of FAPbI₃ (0.95FA and 0.85FA) rather than FAPbI₃ itself. Is there any physical or chemical reason for this? According to the Fröhlich coupling constant, a large strength can be expected if the effective mass of the charge carrier is large and/or the angular frequency of the coupled LO phonon is smaller. Do the authors intend to observe any effects of the substitution on the electron-lattice coupling? Or are there any DFT simulations that indicate any difference in the effective mass or LO phonons by the substitution modification? I am interested in the case where FAPbI₃ is used for the same experiment. The result would be a good reference.

Response: Thanks for the valuable comments. The primary concern of selecting the samples is that (FAPbI₃)_{0.85}(MAPbBr₃)_{0.15} and (FAPbI₃)_{0.95}(MAPbI₃)_{0.05} are the most typical ratio and might be the most stabilized ones in two-step prepared perovskite solar cells. [Nat. Photonics 2019, 13 (7), 500; Adv. Mater. 2022, 34 (8), 2106118; Materials 2021, 14 (14), 4054; J. Am. Chem. Soc. 2015, 137 (51); Adv. Mater. 2017, 29 (35), 1606555; Energy Environ. Sci. 2017, 10 (2), 621]. Therefore, the investigation of polarons in these systems are much more desired than other ratios, and we compared the polaron dynamics in (FAPbI₃)_{0.95}(MAPbI₃)_{0.05} and (FAPbI₃)_{0.85}(MAPbBr₃)_{0.15} in our initial study.

According to your comment, in order to investigate the effects of the substitution on the electron-lattice coupling, we have done THz emission spectroscopy under the same excitation condition in four different samples, FAPbI₃, (FAPbI₃)_{0.95}(MAPbI₃)_{0.05}, (FAPbI₃)_{0.85}(MAPbI₃)_{0.15} and MAPbI₃, as shown in Fig. R7. The terahertz emission at P1 and P2 polaron modes are induced by the anharmonic coupling between the photo-induced carriers and the ionic lattice. Therefore, the emission amplitude at P1 and P2 could represent the strength of electron lattice coupling. As shown in Fig. R7, the terahertz emission at P1 does not change obviously with the A-cation substitution, while the amplitude of P2 mode increased vividly with the MA ratio. This indicates the anharmonic coupling strength between the hot carriers and the A cation rotation mode (P2 phonon mode) would be dependent on the A cation substitution, while the electron

phonon coupling between the carriers and the I-Pb-I angular distortions (P1 phonon mode) would not. It has been reported that the effective mass of electron in FAPbI₃ and MAPbI₃ is 0.095 m_e* and 0.105 m_e* near the band edge of CB1, respectively [Nano convergence, 2016, 3(1): 1-13]. Therefore, the slight P1 mode amplitude discrepancy is consistent with the slight difference of the effective electron mass and is possible due to the small influence of the A cation substitution on the orbital states from the Pb and I occupied the CB1 band edge of both FAPbI₃ and MAPbI₃ [Science, 375(6583), eabj1186; Chem. Rev. 120,7867–7918 (2020)]. In contrast, the effective mass of the hot carrier would increase in the CB2 band at the Γ valley of MAPbI₃ due to the non-parabolicity of the bands away from the band extrema [Phys. Rev. B 2016, 94, 075206; Adv. Mater. 2018, 30, 170731]. According to the Fröhlich coupling constant, a large coupling strength can be expected if the effective mass of the charge carrier is large. Therefore, the electron phonon coupling strength of the P2 mode dependent on the MA ratio is possibly due to the larger effective mass of the hot carriers in CB2 of MAPbI₃. It is also in consistent with the higher dipole moment of MA cation, through which the stronger carrier-molecular dipole interaction could be realized in MA than FA in a cubic phase perovskite lattice at room-temperature [Science, 2016, 353(6306): 1409-1413; Accounts of chemical research 49, 528-535 (2016)]. **For reference comparison, the THz emission spectra of FAPbI₃, (FAPbI₃)_{0.95}(MAPbI₃)_{0.05}, (FAPbI₃)_{0.85}(MAPbI₃)_{0.15} and MAPbI₃ are added to our revised Supplementary Information.**

Figure R7. THz emission spectra of FAPbI₃, (FAPbI₃)_{0.95}(MAPbI₃)_{0.05}, (FAPbI₃)_{0.85}(MAPbI₃)_{0.15} and MAPbI₃, measured with carrier concentration $n=2 \times 10^{18} \text{ cm}^{-3}$.

2. A model based on FAPbI₃ is presented in the manuscript (Fig 2 (d)). I wonder how this model (vertical electronic excitations CB1 and CB2) can be justified under the existence of MAPbI₃. If the authors simply assume this, then is there any DFT simulation method that can support this model? There are several ways to perform DFT simulations to justify this type of model: for instance, the finite size of the structure can be used to calculate both the electronic ground state and the electronically excited state energy along a relevant I-Pb-I normal coordinate (or a simple motion) to see if the equilibrium structure along this coordinate changes. Or the orientation of the FA+ molecule can be altered along with the corresponding normal coordinate (or a simple motion) and the electronic energy can be simulated at each configuration. If the authors can provide these pieces of information, this model can be more convincing.

Response: Thanks for the valuable comments. In the transition model, we assume that electrons are vertically excited from the VB to CB1 and CB2 under the existence of MAPbI₃. According to your suggestion, to make this model more convincing, we

checked the electronic energy curves (Fig. R8) and electronic structure (Fig. R9) associated with the orientation of the FA⁺ molecule. Fig. R8 (b-d) shows the potential energy difference (ΔE) curves of FA rotation along X, Y, and Z axis. When FA rotates or vibrates in a certain direction, the energy difference of the system is less than 0.1eV (depending on the degree of the rotation). This indicates that the orientation of FA cations does not significantly affect the total electronic energy of FAPbI₃. Since the energy difference of the system shown in Fig. R8 varies most along the Y-axis, we also calculated the projected bands structure of FAPbI₃ when FA⁺ rotates 0°, 26° and 90° along the Y-axis (Fig.R9(a-c)). According to our calculation as shown in Fig.R9, the energy of the FA level changes little along K-Path, which means the rotation distortion of the FA⁺ would not change the vertically transition process presented in the Fig. 2.

Fig.R8 (a) Top view of initial atomic structure of FAPI₃, the inset (left down) shows the direction of *a* (*X*) *b* (*Y*), and *c* (*Z*); (b), (c), and (d) Potential energy curves for rotations of FA along X, Y, and Z axis, respectively.

The red-marked energy curves are mainly contributed by FA cations. The calculated fundamental band gap is equal to the optical band gap (the minimum photon

energy required to create an electron-hole pair) plus the exciton binding energy (the energy required to separate the exciton). Fig.R9 shows that the fundamental band gap of VB-CB1 is $\sim 1.5\text{eV}$, which is close our experimental result. The fundamental band gap of VB-CB2 is $\sim 3.1\text{eV}$, which is larger than our experimental value in THz emission spectrum. This is due to the exciton binding energy is not included in our projected band structure calculation. The exciton binding energy of CB1 is 17meV of FAPbI_3 [The Journal of Physical Chemistry Letters 9.3 (2018): 620-627], while the exciton binding energy of organic molecules is considered to be dependent on the molecule length [Applied Physics A, 2003, 77(5): 623-626]. The exciton binding energy of FA could be roughly determined in the range of $0.6\sim 1.3\text{eV}$ based on its radius of 2.89 \AA [Energy Environ. Sci., 2016, 9, 1989-1997; Applied Physics A, 2003, 77(5): 623-626], and the corresponding VB-CB2 band gap is in the range of $1.8\text{eV}\sim 2.5\text{eV}$. Therefore, the vertically transition model based on FAPbI_3 has been confirmed.

Fig. R9 The projected band structure of FAPbI_3 with FA^+ rotates 0° (a), 26° (b) and 90° (c) along the Y-axis, respectively. The red bands are mainly contributed from FA cations.

To further evaluate the coupling model under the existence of MAPbI_3 , we also calculated the projected density of state (PDOS) of FAPbI_3 with and without MA substitution. As shown in Fig. R10(a), the onset of the conduction band of FA cations (Red curve) is about 2.5 eV , which is consistent with previous reports [New Journal of Chemistry 45, 4393-4400 (2021); Acta Phys. Chim. Sin. 2019, 35 (1), 69–75; Scientific reports 11.1 (2021): 1-10]. When doped with MA (only one MA substitutes one FA in a $2 \times 2 \times 2$ supercell, i.e. $\text{FA}_{0.875}\text{MA}_{0.125}$), the shapes of the onset of red curves are similar. Hence, with the existence of MA, the transition process could also be vertical.

We have added the corresponding descriptions in the Supplementary Information S13 of our revised manuscript.

Fig.R10 (a) and (b) Projected density of state (PDOS) of FAPbI₃ and MA doped FAPbI₃, respectively.

3. Electronic excitation can take place at any K -points within the allowed energy window creating electrons and electron holes with each pair of electron-hole having opposite crystal momenta. How is the crystal momentum of phonon involved in Fig. 2(d)? Do the authors assume that THz emission does not need any crystal momentum of the relevant THz phonon? Or if the authors focus only on the Gamma point or specific k -point, please mention it in the manuscript.

Response: Thanks for the valuable comments. The pump light induces excitation of electrons and holes with each pair of electron-hole having opposite crystal momenta. In order to simplify our model, we only draw the electron transition and relaxation process in Fig. 2(d). As discussed in Response 2, the phonons are not involved in the vertical transition process, while the electron-phonon interaction mainly occurs in the cooling process of hot carriers. The contribution of phonons to the crystal momenta is drawn in Fig. R11. According to our proposed anharmonic coupling model, the anharmonic coupling between photo-induced carriers and ionic lattice could lead to bremsstrahlung associated with terahertz phonon, resulting in carriers relaxing into metastable states (polarons) through interaction with phonons in momentum space. Generally, the crystal momenta of the relevant phonons are considered to be related to

the electron phonon coupling process [Nature Reviews Materials 2021, 6(7), 560-586]. Therefore, the crystal momentum of the relevant THz phonon is needed for the THz emission process. **We have mentioned these statements in our revised manuscript in Page 7.**

Fig.R11 Transition model of electron in FAPbI₃, CB1 corresponds to PB-I energy level, CB2 corresponds to FA⁺ energy level, THz emission is induced by anharmonic strong coupling process of the carriers and phonons.

4. Related to comments 1 & 2, can the authors identify the origins of P1 and P2 peaks of 0.95FA are the same as those of MAPbI₃ or FA_{0.83}Cs_{0.17}PbBr_{0.51}I_{2.48} or FAPbI₃? Is it possible for the authors to provide more specific information on the relevant THz modes of 0.95FA (and 0.85FA) and the effects of substitution modification on the modes?

Response: Thanks for the valuable comments. As mentioned in Response 1, we have performed THz emission experiments on the FAPbI₃, (FAPbI₃)_{0.95}(MAPbI₃)_{0.05}, (FAPbI₃)_{0.85}(MAPbI₃)_{0.15} and MAPbI₃, to check the effects of substitution effect on the P1 and P2 polaron modes. As shown in Fig. R7, the terahertz emission at P1 does not change obviously with the A-cation substitution, while the amplitude of P2 mode increased vividly with the MA ratio. This indicates the anharmonic coupling strength between the hot carriers and the A cation rotation mode (P2 phonon mode) would be dependent on the A cation substitution, while the electron phonon coupling between the carriers and the I-Pb-I angular distortions (P1 phonon mode) would not.

Figure R12. (a) The THz conductivity $\sigma(\omega)$ of unexcited 0.95FA film; (b) Real part of transient optical conductivity of 0.95FA with delay time of 5ps and 100ps with pump fluences of $70.3 \mu\text{J}/\text{cm}^{-2}$ at 300 K.

To provide more specific information on the relevant THz modes of 0.95FA. We first investigated the THz time domain spectroscopy (TDS) of the 0.95FA film, as shown in Fig. R12(a). The THz conductivity $\sigma(\omega)$ of 0.95 FA film without optical pump shows the phonon resonance mode at ~ 0.9 THz, which are assigned to the I-Pb-I angular distortions of the inorganic sublattice (P1) [Advanced Materials 30, 1704737 (2018); Science advances 3, e1701217 (2017)]. Then, the optical pump-THz probe (OPTP) spectroscopy of the perovskite film is performed, which could also help to detect the photoinduced carrier coupled phonon modes in the 0.95FA film. [Physical Review Letters, 2019, 122(16): 166601; Light: Science & Applications, 2022, 11(1): 1-12; Nano Lett. 17, 5402(2017)]. As shown in Fig. R12(b), transient optical conductivity acquired at 5ps and 100ps delay times after the photoexcitation are presented. Two resonance peaks at ~ 0.4 THz and ~ 1 THz could be found, which are corresponding to the A-site cation rotation polaron (P2) and inorganic sublattice vibration polaron (P1), respectively. [Journal of the American Chemical Society 139, 4068-4074 (2017); Advanced Materials 30, 1704737 (2018); Light: Science & Applications, 2022, 11(1): 1-12; Science advances 3, e1701217 (2017)] The measured polaron frequencies in OPTP are in consistent with our results obtained using terahertz emission spectroscopy. In contrast, the terahertz emission could present the dynamical anharmonic coupling between the photoinduced carriers and the phonon modes during the formation process

of polarons, which have been seldom reported in HOIPs. Therefore, these measurements of TDS and OPTP could support our new proposed anharmonic coupling model about the strong electron phonon coupling, which is experimentally proved by our terahertz emission spectroscopy. **We have added corresponding descriptions in the revised manuscript. (Page 8)**

REVIEWER COMMENTS

Reviewer #1 (Remarks to the Author):

The authors have addressed my questions and I have no more comments.

Reviewer #2 (Remarks to the Author):

I would like to thank the authors' responses to my comments. The response to my comment is appropriate. I recommend this manuscript can be accepted after the following minor comments are addressed.

- (1) In Fig.1 (a), the Φ is the azimuth angle of the sample, however, it also shown in line 108 for azimuth angle of the polarizer, please clarify.
- (2) The y axis of the inset of Fig. 1 (d) is missing.
- (3) In Fig. 2 (b), the pump photon energy should be mentioned.
- (4) I suggest that Fig. 2 (f) and Fig.3 can be merged into one Figures, as they present the similar information. In addition, why the photoexcited energy 2.06,2.38,2.43,2.58 eV, are not same with the ENERGY axis in Fig. 3(b), please clarify.
- (5) The literation citation should be rechecked. For example, the Ref 11,20,29, 42 have wrong page number.

Reviewer #3 (Remarks to the Author):

The authors' responses to my questions are sufficient and adequate. Now I am satisfied with the revised manuscript. Thus I recommend this version of the manuscript be accepted for publication.

Response to The Reviewers' Concerns:

Reviewer #1:

The authors have addressed my questions and I have no more comments.

Response: We thank the reviewer for providing the constructive comments to improve the quality and accuracy for this manuscript.

Reviewer #2:

I would like to thank the authors' responses to my comments. The response to my comment is appropriate. I recommend this manuscript can be accepted after the following minor comments are addressed.

Response: We thank the reviewer for the positive feedback and valuable suggestions to improve the quality of our work. According to the reviewer's comments, the manuscript has been revised below.

(1) In Fig.1 (a), the Φ is the azimuth angle of the sample, however, it also shown in line 108 for azimuth angle of the polarizer, please clarify.

Response: We thank the reviewer for pointing this out. **We have used “ φ ” to represent the angle of the THz polarizer in our revised manuscript.**

(2) The y axis of the inset of Fig. 1 (d) is missing.

Response: Thanks for the valuable comments. We have added the y axis of the inset of Fig. 1 (d).

(3) In Fig. 2 (b), the pump photon energy should be mentioned.

Response: Thanks for the valuable comments. We have added the information of the pump photon energy (2.58eV) in the caption of Fig. 2(b) in revised manuscript.

(4) I suggest that Fig. 2 (f) and Fig.3 can be merged into one Figures, as they present the similar information. In addition, why the photoexcited energy 2.06,2.38,2.43,2.58 eV, are not same with the ENERGY axis in Fig. 3(b), please

clarify.

Response: Thanks for the valuable comments. We have merged them according to your suggestion, and optimized the layout of Fig.2 and Fig.3. The Energy axis in previous Fig.3(b) only contains the photoexcited energy above 2.4eV, because we have focused on the P2 peak after the collision of the A-site cation with the excited hot carrier. According to your suggestion, we have made Fig.3(c) consistent with the revised Figs.3(a) and (b).

(5) The literation citation should be rechecked. For example, the Ref 11,20,29, 42 have wrong page number.

Response: Thanks for the valuable comments. We have corrected them.

Reviewer #3:

The authors' responses to my questions are sufficient and adequate. Now I am satisfied with the revised manuscript. Thus I recommend this version of the manuscript be accepted for publication

Response: We thank the reviewer for providing very useful comments to help bringing our manuscript to the current level.